# Instant diagnosis of gastroscopic biopsy via deep-learned single-shot femtosecond stimulated Raman histology

Zhijie Liu[1,10], Wei Su[2,10], Jianpeng Ao[1,10], Min Wang[3,10], Qiuli Jiang[4], Jie He[4], Hua Gao[4], Shu Lei[5], Jinshan Nie[6], Xuefeng Yan[7], Xiaojing Guo[8], Pinghong Zhou[2✉], Hao Hu[2,9✉] & Minbiao Ji[1✉]

Gastroscopic biopsy provides the only effective method for gastric cancer diagnosis, but the gold standard histopathology is time-consuming and incompatible with gastroscopy. Conventional stimulated Raman scattering (SRS) microscopy has shown promise in label-free diagnosis on human tissues, yet it requires the tuning of picosecond lasers to achieve chemical specificity at the cost of time and complexity. Here, we demonstrate that single-shot femtosecond SRS (femto-SRS) reaches the maximum speed and sensitivity with preserved chemical resolution by integrating with U-Net. Fresh gastroscopic biopsy is imaged in <60 s, revealing essential histoarchitectural hallmarks perfectly agreed with standard histopathology. Moreover, a diagnostic neural network (CNN) is constructed based on images from 279 patients that predicts gastric cancer with accuracy >96%. We further demonstrate semantic segmentation of intratumor heterogeneity and evaluation of resection margins of endoscopic submucosal dissection (ESD) tissues to simulate rapid and automated intraoperative diagnosis. Our method holds potential for synchronizing gastroscopy and histopathological diagnosis.

[1] State Key Laboratory of Surface Physics and Department of Physics, Human Phenome Institute, Academy for Engineering and Technology, Key Laboratory of Micro and Nano Photonic Structures (Ministry of Education), School of Life Sciences, Yiwu Research Institute, Fudan University, 200433 Shanghai, China. [2] Endoscopy Center and Endoscopy Research Institute, Shanghai Collaborative Innovation Center of Endoscopy, Zhongshan Hospital, Fudan University, 200032 Shanghai, China. [3] Department of Gastroenterology, Shanghai Children Hospital, Shanghai Jiaotong University, 200062 Shanghai, China. [4] Department of Pathology, Endoscopic Center, Zhongshan Hospital (Xiamen Branch), Fudan University, 361015 Xiamen, China. [5] Department of Gastroenterology, Wuhan No. 1 Hospital, 430014 Wuhan, China. [6] Department of Gastroenterology, the 1st People's Hospital of Taicang, Soochow University, 215400 Suzhou, China. [7] Department of Gastroenterology, Shangrao Municipal Hospital, Shangrao 334000 Jiangxi, China. [8] Department of Health Statistics, Faculty of Health Service, Naval Medical University, Shanghai, China. [9] Department of Gastroenterology, People's Hospital of Shigatse, 857007 Shigatse, China. [10] These authors contributed equally: Zhijie Liu, Wei Su, Jianpeng Ao, Min Wang. ✉email: zhou.pinghong@zs-hospital.sh.cn; hu.hao1@zs-hospital.sh.cn; minbiaoj@fudan.edu.cn

Gastric cancer (GC) is one of the most prevalent malignant carcinomas worldwide, and gastroscopy is considered to be the most effective screening method[1]. Pathological examination of gastroscopic biopsy is the only method for precise GC diagnosis, determining treatment strategies and patient prognoses[2]. However, gold standard staining-based pathological workflow is incompatible with gastroscopy, due to the labor- and time- intensive processes including fixation, sectioning and staining[3]. Moreover, traditional histological methods require well-trained pathologists to interpret tissue slides, which is not only impractical for timely intraoperative consultation, but also constrained by the wide variation among pathologists in the diagnosis and grading of tumor subtypes. Therefore, advanced technique for rapid and automated intraoperative histopathology of gastroscopic biopsy and resection specimen is essential to improve patient treatments and surgical outcomes.

Stimulated Raman scattering (SRS) microscopy is a fast-growing technology with the capabilities of rapid and chemical-specific imaging of native biomolecules based on their fingerprint vibrational spectra[4–13]. These advantages have enabled label-free histopathology of fresh tissue specimens, revealing key diagnostic histological features of various types of human solid tumors and diseases[14–21]. Stimulated Raman histology (SRH) images the distributions of lipids and protein selectively with picosecond laser pulses to provide sufficient spectral/chemical resolution, using either narrow band sources[14,15,18,21,22], or broadband chirped pulses[23–25]. However, multi-color picosecond SRS (pico-SRS) requires the tuning of wavelength/time-delay or complex optical engineering[8,16,26–29], sacrificing sensitivity and imaging speed (several minutes on ~$2 \times 2$ mm$^2$ specimens). In contrast, femtosecond SRS (femto-SRS) generates a single image with much higher signal-to-noise ratio (SNR) and faster imaging speed because of the greater pulse peak power, but is hardly used directly because it lacks spectral resolution for chemical imaging[30,31]. Recent advances in deep-learning neural networks have enabled the extraction of spatially correlated spectra from femto-SRS images by training with picosecond hyperspectral SRS data[32,33], but the full spectral information is redundant for histological imaging and inefficient for intraoperative diagnosis.

Here, we leveraged the advantage of U-Net based neural network to project the single-shot femto-SRS image to the dual-channel picosecond ones with recovered biomolecular contrast and sub-minute imaging time. Femto-SRH of fresh gastroscopic biopsy successfully revealed the same histological hallmarks as traditional hematoxylin and eosin (H&E) staining. Moreover, a second deep-learning architecture based on convolutional neural network (CNN) was developed for the classification of non-cancer, differentiated cancer and undifferentiated cancer, yielding high diagnostic accuracy and concordance with conventional H&E. Finally, semantic segmentation was realized on femto-SRH to visualize the heterogeneous distribution of tumor subtypes within a biopsy, which was further used to evaluate resection margins of endoscopic submucosal dissection (ESD) tissues to simulate intraoperative diagnosis.

## Results

**Experimental design and dual-mode SRS microscope**. The goal of our study was to verify the possibility of prompt histologic imaging on fresh gastroscopic biopsy using femto-SRS microscope (Fig. 1a). To fully exploit the advantages of femto-SRS (high speed and SNR) and circumvent its inability in chemical selectivity, we constructed a dual-mode SRS microscope, capable of imaging the same FOVs with femto- and pico-SRS (Fig. S1). The optical path of femto-SRS simply uses the transform-limited femtosecond pulses (~120 fs) to generate single-channel SRS images at zero time-delay without chemical resolution. In contrast, spectral-focusing based pico-SRS requires the chirping of femtosecond pulses to ~2–4 picoseconds through dispersive optics, yielding SRS images at different Raman frequencies by varying the inter-pulse time delays (Fig. 1b). Traditional pico-SRS takes raw images at two Raman frequencies ($\omega_1 = 2845$ cm$^{-1}$ for CH$_2$, $\omega_2 = 2930$ cm$^{-1}$ for CH$_3$) to extract lipid/protein distributions and generate histological images. The two SRS modes share most of the optical components except the pulse stretching parts, and could be switched by shutters to image the same FOVs without noticeable sample shifts. Therefore, large datasets of the corresponding femto- and pico-SRS images could be harvested to train the deep neural networks, so that single-shot femto-SRS could be converted to dual-channel pico-SRS without complex optical engineering or physical tuning of detection frequencies. Moreover, imaging speed could be doubled with about half the laser power. We also include second harmonic generation (SHG) of collagen fibers to generate composite multi-color SRS images, as well as false-colored SRH images to reveal histological features and for further analysis (Fig. 1c).

**Chemical imaging via U-Net based Femto-SRS**. The chemical resolution of SRS essentially results from its spectral resolution to distinguish molecules of different vibrational spectra. Note that femto-SRS integrates the spectral-domain information into a single image, whereas pico-SRS generates hyperspectral images dispersed in the spectral or time domain (Fig. 1b). Therefore, it is the inverse problem of recovering discrete spectral images from the summed image by deep learning algorithms.

We applied a U-shaped fully convolutional network (U-Net) to project single-shot femto-SRS images to dual-channel pico-SRS images with preserved spatio-chemical information. Live HeLa cells and fresh gastric tissues were imaged with both femto- and pico-SRS on the same FOVs. The differences between the two imaging modes could be clearly visualized. For instance, in live cell images pico-SRS revealed high intensity of CH$_3$ (2930 cm$^{-1}$) and low intensity of CH$_2$ (2845 cm$^{-1}$) vibrations in the protein-rich cell nucleus, while lipid-rich organelles and droplets showed high intensities at both Raman frequencies (Fig. 2a). These are known from the spectral differences between lipids and protein[14]. In strong contrast, femto-SRS offered a single image with integrated spectral intensity within the CH stretch window, including the two Raman channels imaged with pico-SRS.

For the training of U-Net, 50 FOVs of HeLa cells and 100 FOVs of gastric tissues were used, with the ratio of training and test data size set to 4:1. After training and optimization, the U-Net was able to convert the raw femto-SRS image to dual-channel images that resembled the pico-SRS at the corresponding Raman frequencies. The high conversion accuracy could be seen by the similarity between the measured pico-SRS (ground truth) and U-Net predicted femto-SRS images of the same Raman frequencies (Fig. 2a). The intensity profiles along the line-cuts through the live cell images further confirmed the accuracy of U-Net conversion, as well as the recovered chemical contrast as highlighted in the nucleus regions (Fig. 2b). The dual-channel SRS images were processed to form colored composite images of cells and tissues, demonstrating almost identical results between pico-SRS and deep learned femto-SRS (Fig. 2c, d and Fig. S2).

**Processing of femto-SRH images**. After U-Net conversion into dual-channel SRS images, chemical decomposition was applied to yield the images of lipids and protein by simple linear algebra[14]. Collagen fibers were represented by SHG signal directly. In traditional multi-color SRS imaging, collagen, lipid and protein are false-colored red, green and blue, respectively (Fig. 2c, d).

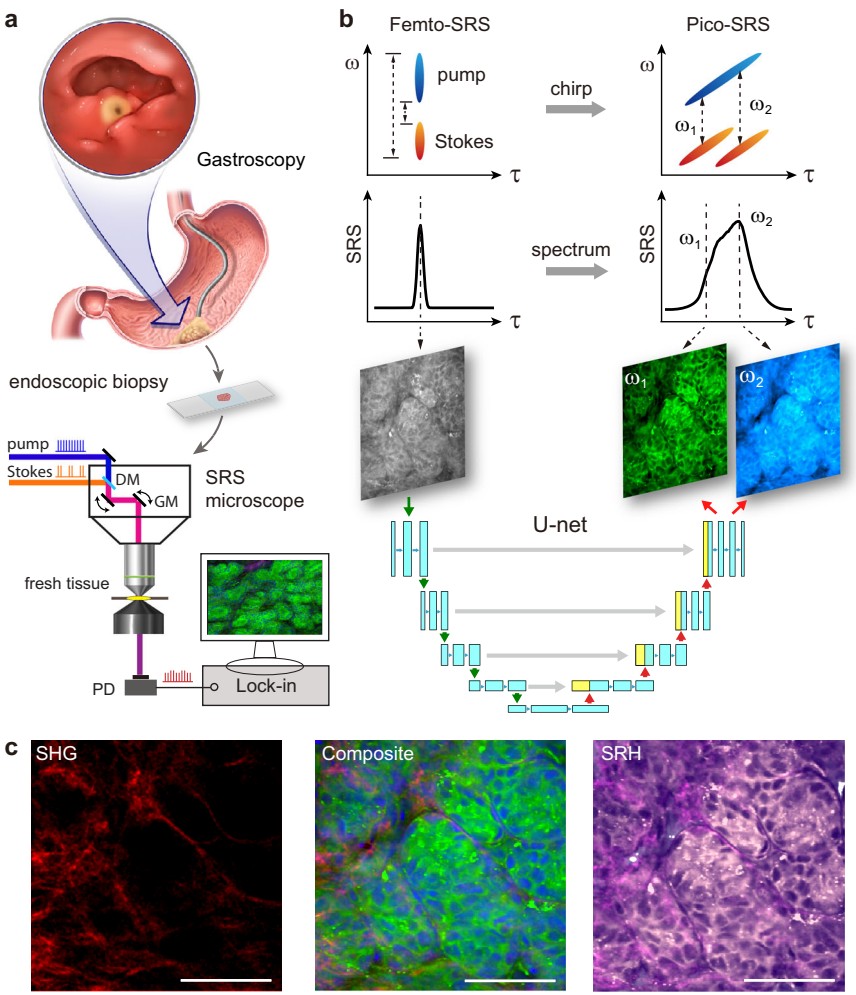

**Fig. 1 Experimental design and workflow. a** Illustration of gastroscopy and taking fresh biopsy for direct SRS imaging. **b** Properties of femto-SRS and pico-SRS, in terms of pulse chirping, spectral resolution, and conversion from single femto-SRS image to a pair of pico-SRS images using deep U-Net. **c** Multi-chemical imaging of gastric tissue composed of lipid (green), protein (blue) and collagen fibers (red) from converted femto-SRS and SHG channels, and color-coded to SRH. DM dichroic mirror, GM galvo mirror, PD photodiode, SRS stimulated Raman scattering, SHG second-harmonic generation, SRH stimulated Raman histology. Scale bars: 50 μm. Source data are provided in the Source data file.

To simulate the coloring scheme of H&E, we created the pseudo-color of SRS to generate SRH that was more familiar to pathologists[18]. The contents of lipid, protein and collagen were projected to light pink, dark purple and magenta, respectively, forming SRH images more akin to traditional histopathology (Fig. 1c and Fig. 3). Although the lipid/protein-based contrast of SRH was not identical to traditional H&E, it revealed important pathological features highly similar to H&E.

**Femto-SRH reveals key diagnostic features of fresh gastric tissues**. Rapid imaging of femto-SRS was achieved on fresh gastroscopic biopsy (~2 × 2 mm²) within 1 min, followed by real-time U-Net conversion and pseudo coloring to generate multi-color femto-SRS and femto-SRH images (Movies S1 and S2). All the important normal and neoplastic gastric tissue histoarchitectures were clearly shown, highly consistent with the findings of standard H&E. The results of typical non-cancerous tissues are shown in Fig. 3. SRS readily showed the regular arrangement of gastric epithelial cells and surrounding basement membrane in normal glands (Fig. 3a). In adenomas, SRS clearly showed mild to moderate dysplasia of the glandular epithelium, with nuclei locating at the base and maintaining polarity, cigar-shaped

features (Fig. 3b). For intestinal metaplasia, in which intestinal-type epithelium replaces normal gastric mucosa, SRS could identify the cup-shaped cells within the metaplastic epithelium, whose secreted mucus was shown as large round compartments with moderate protein content (Fig. 3c, arrows). In addition, SRS also provided a clear view of the extraductal and surrounding mesenchyme. In the case of inflammation, SRS identified infiltration of inflammatory cells within the lamina propria, which was characterized by small, dense, irregularly arranged nuclei (Fig. 3d).

For neoplastic lesions, femto-SRH showed distinct histopathologic features of adenocarcinoma with different degrees of differentiation. In well-differentiated adenocarcinomas, SRS depicted abnormal glands composed of cells with prominent atypia, enlarged nuclei, loss of polarity, and extension toward the luminal surface (Fig. 4a). In moderately differentiated adenocarcinomas, SRS clearly showed disorganized neoplastic cells and distorted ducts, such as irregular, extended, angular, and abnormally fused glandular ducts in tubular adenocarcinomas (Fig. 4b) and elongated finger-like processes lined by columnar or cuboidal cell supported by fibrovascular connective tissue cores in papillary adenocarcinomas (Fig. 4c). As for poorly differentiated adenocarcinoma, SRS showed malignant epithelial cells in a single

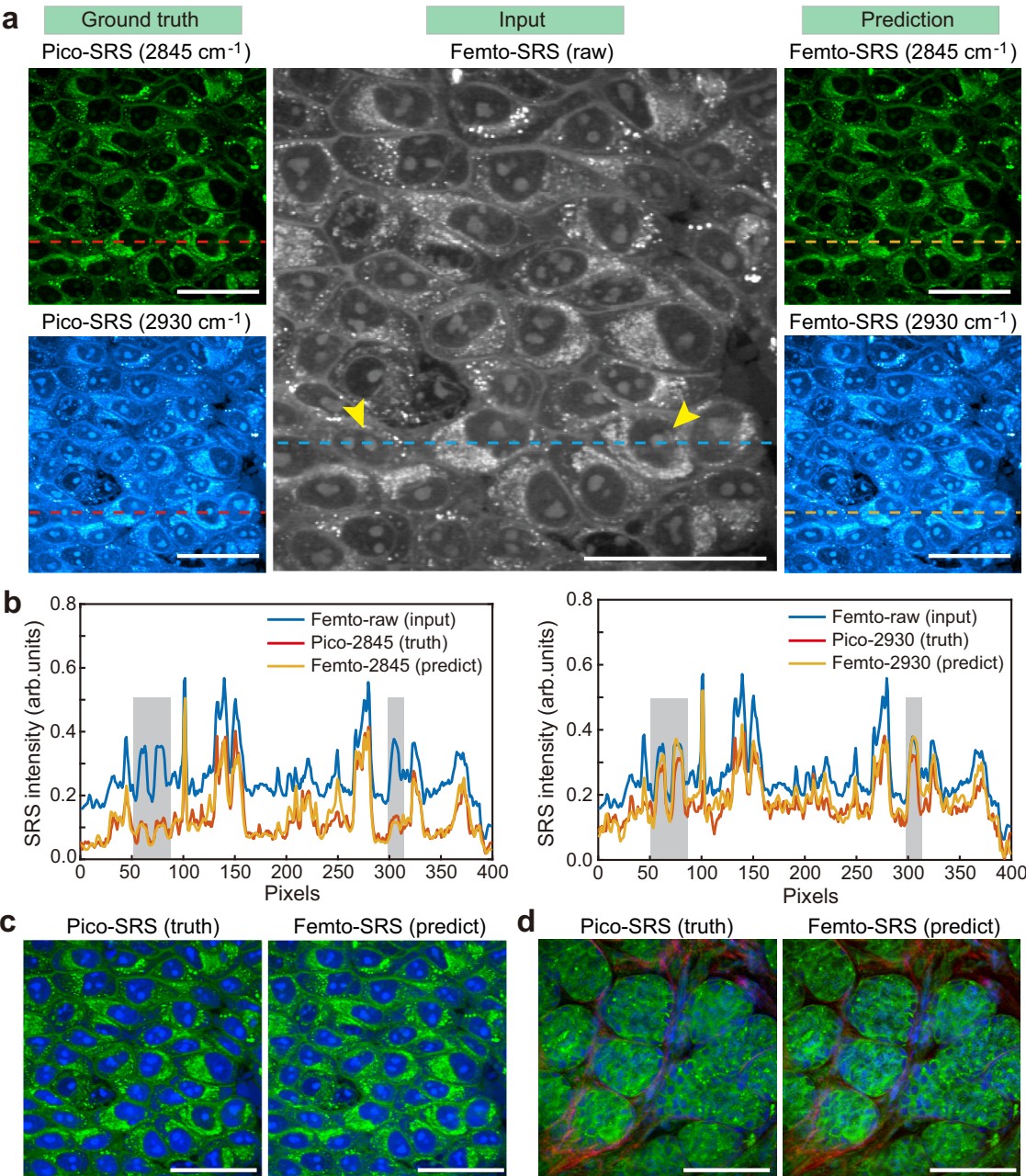

**Fig. 2 Deep U-Net based femto-SRS imaging with recovered chemical resolution. a** Single raw femto-SRS image (middle) was converted by U-Net into the spectrally resolved images at 2845 cm$^{-1}$ and 2930 cm$^{-1}$ (right), compared with the ground truth pico-SRS images (left). **b** Intensity profiles (dashed lines in (**a**)) of the predicted and ground-truth data, showing chemical contrast in the cell nucleus regions (shadows, and arrows in (**a**)). **c, d** Femto-SRS imaging of live cells and gastric tissues, yielding almost identical results to pico-SRS. Green: lipids; blue: protein; red: collagen fibers. Scale bars: 50 μm. Source data are provided in the Source data file.

or nest-like distribution without forming tubular structures (Fig. 4d). In addition, SRS was even able to differentiate signet ring cell components similar to that of H&E staining, showing abundant intracellular mucus pushing the nucleus to one side (Fig. 4e).

Despite the overall high agreement between SRS and H&E, a few discrepancies remain and are noteworthy. First of all, SRS maps out the distributions of lipids and protein, whereas H&E mainly stains nucleic acid and protein. Secondly, we demonstrated SRS images on fresh tissues in comparison with H&E staining on thin sections. Although SRS images create thin optical sections of ~1 μm thickness due to the nonlinear property of optical signal[5,34], they showed slightly different

structural features from that of H&E. For instance, the core of the glandular ducts usually appeared hollow in H&E but filled in SRS (Fig. 3a, b and 4a), which is likely due to the loss and dissolution of contents during the sectioning and deparaffinization processes of H&E. And the dehydration process might result in denser cellular patterns in H&E. Therefore, SRS might reveal more intact and better preserved histoarchitectures of fresh gastric tissues (Fig. S3). Furthermore, label-free SRS is free from the large staining variation of H&E as seen in the different colorings among tissues (Figs. 3 and 4). The more consistent appearance of SRS images may help reduce the uncertainty and improve the accuracy of machine learning based diagnosis.

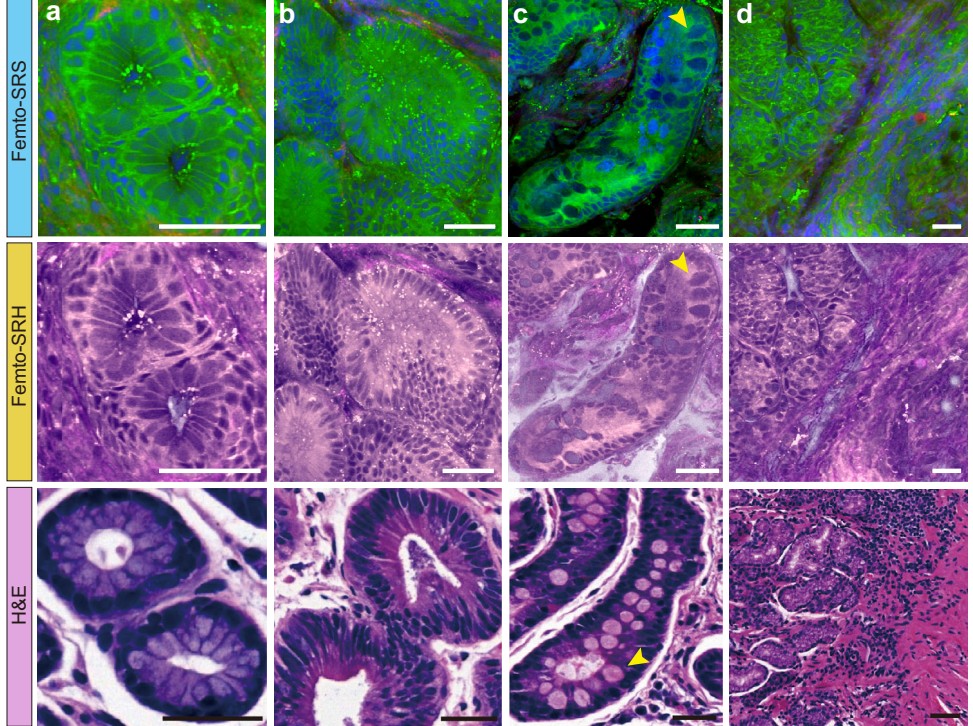

**Fig. 3 Femto-SRS and femto-SRH reveal typical histopathological features of non-cancerous tissues. a** Normal gastric gland; **b** Adenoma; **c** Intestinal metaplasia with the characteristic large cup-shaped cells (arrows); **d** Moderate mucosa inflammation. Green: lipids; blue: protein; red: collagen fibers. Scale bars: 50 μm. Source data are provided in the Source data file.

**Deep-learning assisted diagnosis and classification**. To reduce the workload of pathologists and provide potentially more objective diagnosis for intraoperative histopathology, we evaluated the performance of deep-learning based diagnosis on femto-SRS images from 279 patients (Table S1). The preferred CNN was Inception-ResNet-V2, composed of an inception module and a ResNet module. The CNN was trained to classify the images into three categories: non-cancer, differentiated cancer and undifferentiated cancer. Labeling of the image data for supervised learning was based on the pathology of each gastric biopsy: normal, adenoma, intestinal metaplasia and gastritis cases were labeled as "non-cancer"; among cancerous tissues, well or moderately differentiated tubular and papillary adenocarcinomas were labeled as "differentiated cancer"; poorly differentiated adenocarcinomas, mucinous adenocarcinoma and signet ring cell carcinomas were labeled as "undifferentiated cancer"[35,36]. Two neural networks with binary outputs were used for the classification, the first one categorized the dataset into the non-cancerous and cancerous lesions, and the second one further classified the cancer group into subgroups of differentiated and undifferentiated cancer.

The femto-SRS image data set was divided into training/validation set and test set. The training/validation images were sliced into standard tiles of $300 \times 300$ pixels, and further divided into the training set and validation set with a ratio of 4:1 using fivefold cross validation. The number of cases and tiles for all the data subsets are shown in Table S2. The training input of each CNN was a patch of tiles, and output the prediction value for each tile (Fig. 5a and Fig. S4). The test images were kept independent from the training process, and the subsequent prediction value was made for individual patient cases based on the classification percentage of the whole SRS image with a threshold of ~0.5[16,18], as determined by the Youden's index from the training results (Fig. S5).

For the CNN that classified non-cancer/cancer, the training curves, test results and receiver operating characteristic (ROC) curves demonstrated high performance (Fig. 5b, c and Fig. S5), with the area under the curve (AUC) of 98.8%. Among the 55 test cases, 2 non-cancer cases were mis-classified as cancer (Fig. 5b and Fig. S6). For the network that classified differentiated/undifferentiated cancer, slightly degraded performances are shown (Fig. 5d, e and Fig. S5), with the AUC of 94.2%. And of the 17 test cases, 1 differentiated case was mis-classified as undifferentiated (Fig. 5d and Fig. S6). The overall confusion matrix of the CNN-based femto-SRH in differentiating the three diagnostic categories is shown in Fig. 5f. We then quantitatively evaluated the diagnostic results of the CNN-predictions on femto-SRH images and the rating of four professional pathologists on the H&E sections of the corresponding specimens with the ground truth being the results of clinical pathology (Table 1). Our results indicated that CNN-based SRH has near-perfect diagnostic performance in determining cancerous lesions (96.4% accuracy), and a high level of concordance with the true pathology ($\kappa = 0.899$). The statistical differences between CNN-SRH and the pathologists were subtle in the diagnosis of cancerous lesions (Table S3). For the subtyping of differentiated and undifferentiated cancers, CNN-SRH demonstrated better performance (94.1% accuracy) than the pathologists (85.3% mean accuracy), and maintained a high diagnostic concordance with true pathology ($\kappa = 0.85$). Despite the few errors, our trained CNN models exhibit excellent capability of providing automated and accurate diagnosis on femto-SRH images of fresh gastric tissues.

**Semantic segmentation of intratumoral heterogeneity**. The highly heterogeneous development of tumor results in the uneven intratumoral histoarchitectures, which offers critical information of tumor malignancy. To further visualize the heterogeneous

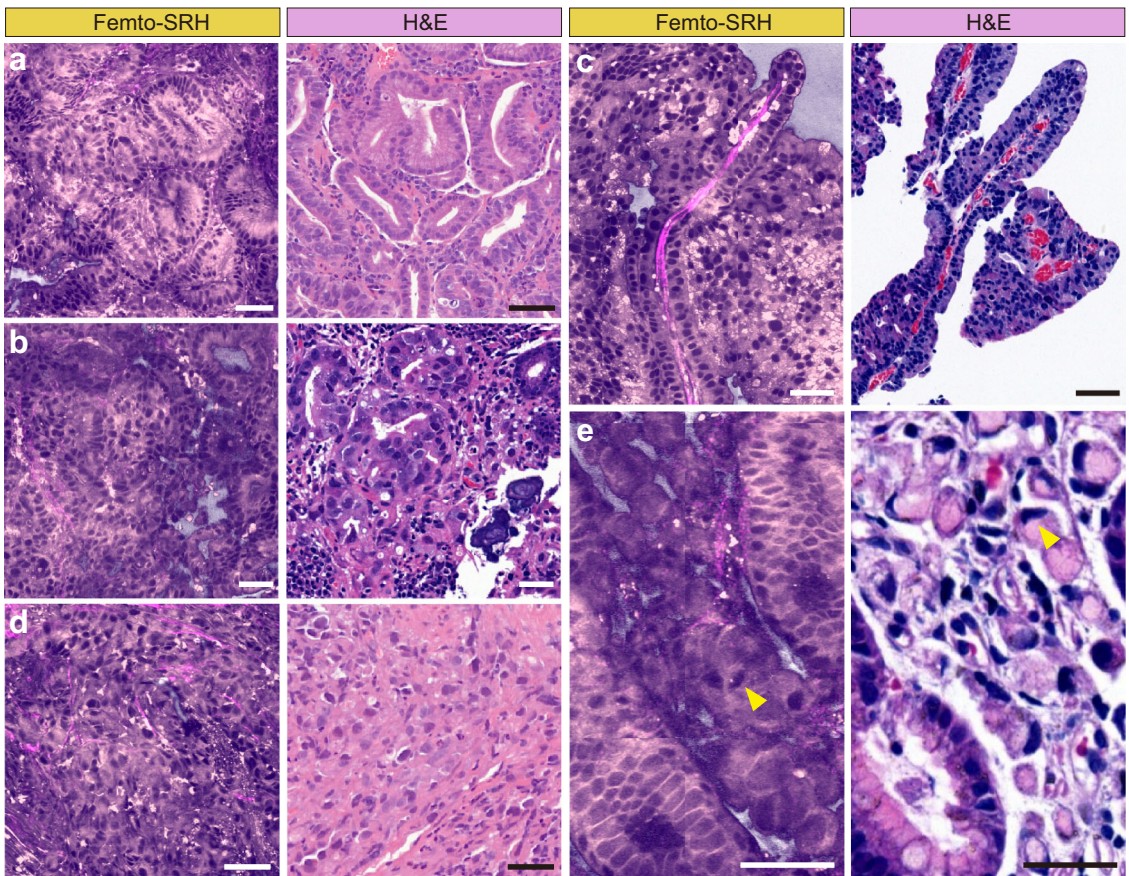

**Fig. 4 Characteristic femto-SRH images of neoplastic fresh gastroscopic biopsies.** Differentiated carcinomas include: (**a**) well-differentiated adenocarcinoma, (**b**) moderately differentiated tubular adenocarcinoma, and (**c**) differentiated papillary adenocarcinoma with fibrovascular connective tissue cores. Undifferentiated carcinomas include: (**d**) poorly differentiated adenocarcinoma, and (**e**) signet ring cell (arrowheads) carcinoma. Scale bars: 50 μm. Source data are provided in the Source data file.

distribution of tumor subtypes and levels of differentiation, we developed a semantic segmentation algorithm based on the above CNNs, with the operations of flip-expansion, shifting FOVs and scoring (see Methods and Fig. S7). High-resolution predictions were made to generate the diagnostic probability maps of non-cancer, non-diagnostic, differentiated and undifferentiated cancers (Fig. 6a). The prediction probabilities were colored and superimposed onto the SRH images, showing the spatial characteristics of the diagnostic histological heterogeneity within a gastroscopic biopsy (Fig. 6b).

To illustrate the performance of the algorithm, we demonstrated the results of a typical tumorous tissue imaged with femto-SRH (Fig. 6b). The red color represents the high probability regions of predicted undifferentiated cancer, with low differentiation, poor maturity and high malignancy. The blue color represents the differentiated tissue regions with well or moderately differentiated carcinoma, and lower degree of malignancy. The green regions are predicted non-cancer, with gastritis symptoms and regularly patterned histology. And the gray areas do not provide useful diagnostic information, mainly consisting of the empty regions. From the segmentation results, we could clearly see that this particular tissue was mainly composed of poorly differentiated tumors with very few sites that remained moderate differentiation, which agreed with the pathological result of the patient. Moreover, representative FOVs of these subtyped areas are magnified (Fig. 6c) and verified by pathologists, confirming the accuracy of the semantic segmentation algorithm in generating diagnoses with high spatial

resolution. Pixel-level prediction accuracy is difficult to achieve at current stage, since corresponding SRH and H&E results could not be generated on fresh tissues.

**Intraoperative evaluation of resection margins for ESD**. In addition to the application of deep learned femto-SRH in rapid diagnosis of fresh gastroscopic biopsy, we also demonstrated its potential in evaluating resection margins for endoscopic submucosa dissection (ESD), whose goal was to achieve maximal removal of tumor tissues and preservation of healthy parts. We simulated intraoperative diagnosis on ESD specimens taken at three locations: the tumor core, the visual margin and the normal tissue ~8 mm away from the margin (Fig. 7a). After resection and non-invasive imaging with femto-SRH, the fresh tissues were subsequently sent to the pathology department for standard histopathological examinations as the ground truths.

The femto-SRH image of each tissue was evaluated based on the diagnostic histological features and semantic segmentation using the deep learning models we have developed (Fig. 7b), highlighting the areas predicted as non-cancer (blue) and cancer (red). For the intra-tumor tissue, SRH revealed well-differentiated tubular adenocarcinoma within most of the tissue areas, well-agreed with the traditional pathology. At the tumor margin, a clear boundary between the normal and tumorous tissues could be identified, with differentiated tubular adenocarcinoma in the predicted tumor region. As for the adjacent normal tissue, the overall tissue histoarchitecture appeared non-cancerous with only

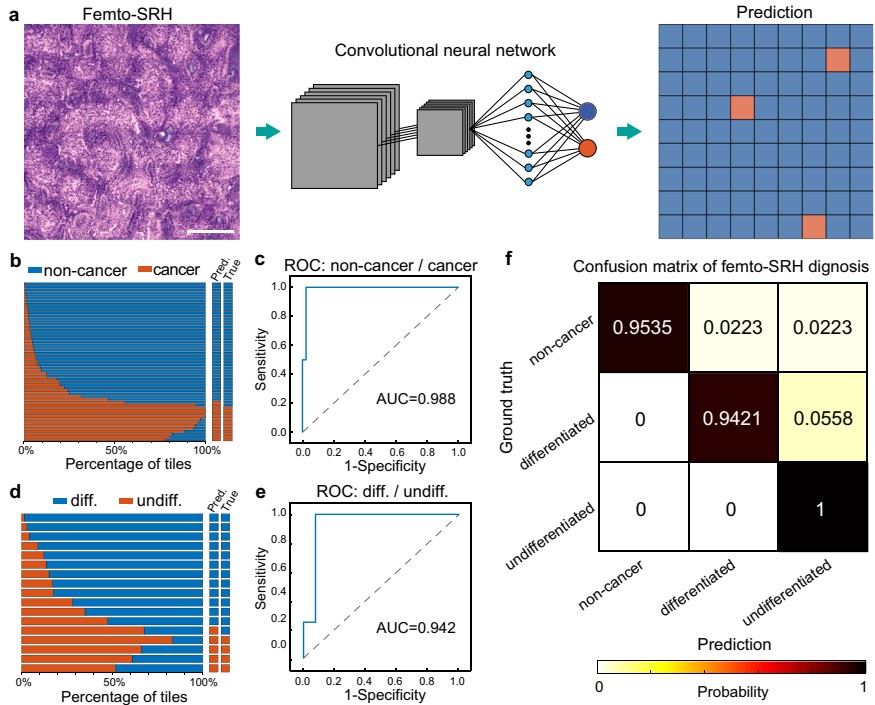

**Fig. 5 Automated diagnosis with deep convolutional neural networks (CNNs) on femto-SRH images. a** Schematics of CNN predictions; **b**, **c** Training, validation and test results of the neural network for the classification of non-cancer/cancer tissues; **d**, **e** Training, validation and test results of the neural network for the classification of differentiated/undifferentiated cancers. **f** Confusion matrix of the three diagnostic subtypes. Scale bar: 200 μm. Source data are provided in the Source data file.

**Table 1 Diagnostic comparison between femto-SRH and pathologists.**

| | SRH | | H&E | | | | | | | | | Combined accuracy (%) |
|---|---|---|---|---|---|---|---|---|---|---|---|---|
| | CNN | | Pathologist 1 | | Pathologist 2 | | Pathologist 3 | | Pathologist 4 | | | |
| | Correct | Wrong | Correct | Wrong | Correct | Wrong | Correct | Wrong | Correct | Wrong | | |
| Non-cancer vs. Cancer | | | | | | | | | | | | |
|   Non-cancer | 41 | 2 | 40 | 3 | 42 | 1 | 42 | 1 | 43 | 0 | | 96.7 |
|   Cancer | 12 | 0 | 11 | 1 | 12 | 0 | 10 | 2 | 11 | 1 | | 93.3 |
| Total | 53 | 2 | 51 | 4 | 54 | 1 | 52 | 3 | 54 | 1 | | 95.9 |
| Accuracy (%) | 96.4 | | 92.7 | | 98.2 | | 94.5 | | 98.2 | | | |
| Concordance | 0.899 | | 0.799 | | 0.948 | | 0.835 | | 0.945 | | | |
| Cancerous lesion | | | | | | | | | | | | |
|   Differentiated | 12 | 1 | 12 | 1 | 11 | 2 | 11 | 2 | 13 | 0 | | 90.8 |
|   Undifferentiated | 4 | 0 | 3 | 1 | 3 | 1 | 3 | 1 | 2 | 2 | | 75 |
| Total | 16 | 1 | 15 | 2 | 14 | 3 | 14 | 3 | 15 | 2 | | 87.1 |
| Accuracy (%) | 94.1 | | 88.2 | | 82.4 | | 82.4 | | 88.2 | | | |
| Concordance | 0.85 | | 0.673 | | 0.549 | | 0.549 | | 0.605 | | | |

a small amount of infiltrated cup-shaped cells, which agreed with the diagnosed chronic atrophic gastritis with intestinal metaplasia. A diagnosis accuracy of ~93.3% was reached among the five studied ESD cases (Fig. 7b). Rapid evaluation of resection margins with femto-SRH may assist the decision-making during ESD and help improve patient care.

## Discussion

In the current study, we have developed a U-Net based femto-SRS imaging method to realize label-free tissue histology with a simple optical layout and improved SNR, without the need to manipulate in the frequency or time domain. The recovered flavors of chemical contrast provide necessary histological details for both human and machine-learning model to reach high diagnostic

accuracy. And the cost of these advances is mainly in the training of the deep neural network, which requires the designed dual-mode system to generate paired femto- and pico-SRS images. In general, the training should be done separately for different types of tissues, including gastric, brain, breast etc., while the dual-mode imaging system could be shared. It could in principle provide the highest imaging speed with the optimum laser scanning schemes, such as resonant glavo-mirrors[5]. And combined with advanced tissue sliding technique[37], $2 \times 2 \, mm^2$ specimens may be imaged with <30 s, which will be a significant improvement of rapid histology to minimize the perturbation on ordinary gastroscopic examination process. The optical simplicity, stability and speed improvement of our method may become more advantageous for clinical translation.

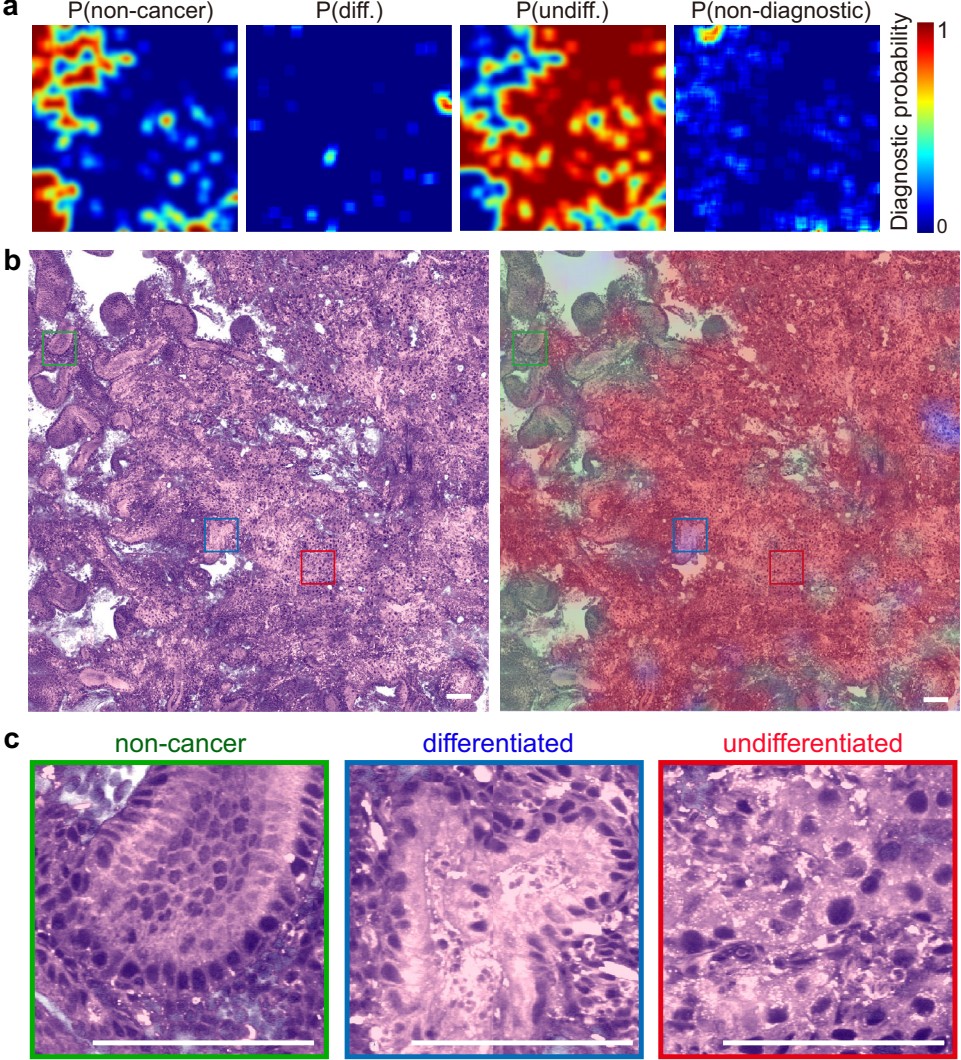

**Fig. 6 Semantic segmentation reveals intratumor heterogeneity. a** Spatial distribution of the prediction probabilities of each category; **b** Femto-SRH image and the segmented distributions of non-cancer (green), differentiated cancer (blue), undifferentiated cancer (red) and non-diagnostic (gray) regions; **c** Magnified view of each category. Scale bars: 100 μm. Source data are provided in the Source data file.

Recent progress of AI-assisted diagnosis based on conventional pathological tissue sections has shown great promise[38–40]. However, the time-consuming preparation steps of conventional sections do not fully meet the needs of endoscopic biopsy and diagnosis. The label-free chemical imaging advantage of SRS provides the unique capability of diagnosing fresh tissues in near real-time. Given the lack of an intraoperative diagnostic tool for gastroscopic biopsy, deep learning based femto-SRH may add a critical component to gastroscopic examinations, providing endoscopists with important references for making treatment decisions, and a critical time window for the patients' psychological preparation. Moreover, the non-invasive imaging method allows the specimens to be reusable for subsequent histopathological tests to avoid duplicated sampling, such as immunohistochemistry, FISH, PCR, etc.

Experienced pathologists from different hospitals were recruited to include diagnostic variation among different interpretations of pathological criteria. Our findings showed that the diagnostic ability of femto-SRH was comparable to that of experienced pathologists in distinguishing cancerous from non-cancerous lesions, but slightly superior in determining the degree of differentiation (Fig. S6). Two undifferentiated cases were

incorrectly diagnosed as differentiated by pathologists (patients #106 and #120), and three differentiated cases were incorrectly rated as undifferentiated (patients #57, #58 and #164). In contrast, CNN based femto-SRH showed better agreement with ground truth. These results indicate that femto-SRH may provide more objective and uniform diagnostic criteria to minimize variation among pathologists due to differences in the training experience[21]. Moreover, imaging of fresh specimens could avoid tissue processing artifacts and provide more consistent and intact tissue histoarchitectures for improved diagnosis. For the two misclassified non-cancer cases by CNN (Fig. S8), we found the tiles with tissue degradation (#54) or inflammation with dense macrophage-like cells (#186) were more likely to be mis-predicted as cancer. We also noticed that the slightly more collagen fibrosis tended to appear in cancer cases than non-cancer ones (Fig. S9). Although collagen fibers are known to correlate with tumor structures and metastasis[41–43], in our CNN model, the collagen channel was found to have minimal effect on prediction accuracy, in accord with that collagen was not the key histological feature for cancer diagnosis.

The diagnosis of progressive gastric cancer could also be realized with conventional white-light endoscopy (WLE) by the

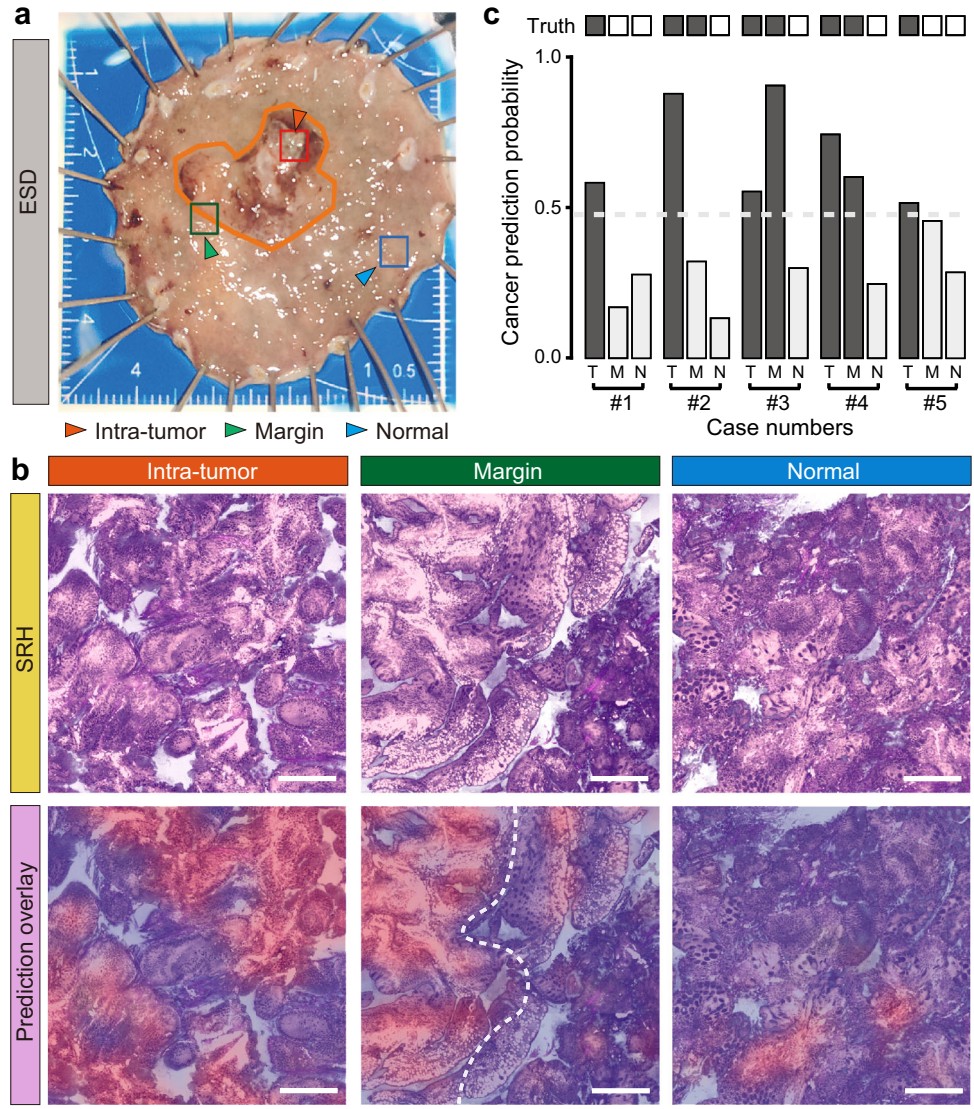

**Fig. 7 Simulated intraoperative evaluation of resection margins on endoscopic submucosal dissection (ESD) specimens. a** A typical ESD tissue and the three test locations of intra-tumor (T), visual margin (M) and ~8 mm away from the margin (N); **b** Femto-SRH imaging results of the three test points of a typical case, showing the predicted diagnostic segmentations of cancer (red) vs. non-cancer (blue). **c** SRH prediction results of the five ESD cases compared with ground truths. Scale bars: 200 μm. Source data are provided in the Source data file.

evident morphologically features. However, the diagnosis of early gastric cancer (EGC) is challenging. To evaluate the performance of femto-SRH, we tested a total of 15 samples from 3 different sites on 5 ESD specimens with nearly 100% accuracy using CNN based femto-SRH. It can successfully display the cancerous and non-cancerous areas and delineate the boundaries between them, which is essential for curative endoscopic resection. Although magnifying endoscopy with narrow-band imaging (ME-NBI) has been applied to preoperatively delineate the endoscopic resection margins of EGC, it was found to be inaccurate in 12% of EGC patients[44]. Our future work may evaluate the performance of femto-SRH in delineating the EGC cases in comparison with ME-NBI.

Aside from the demonstrated capabilities, our method could be further improved in several aspects. First, the current CNN prediction was based on a percentage threshold of the small tiles, which differed from the pathologists' criteria with only a few cancer sites identified on a slide to judge a cancer case. More advanced deep-learning models (such as the weakly-supervised learnings) will be needed to reduce labeling workload, weaken labeling errors, and improve the accuracy of identifying key diagnostic sites. Second, the SRH coloring performance relies on the biochemical compositions of tissues, such as lipid/protein ratio and collagen content[45,46]. In gastric tissues, we found quite consistent chemical compositions among all the cases studied (Fig. S9), ensuring consistent SRH qualities. Future work will include deep-learning based SRH to automatically generate images more akin to H&E. Third, we had not included rare types of gastric cancer to develop deep femto-SRH because of the limited cases and data size. Increasing the number of the rare types of gastric cancer would help to refine the networks for classifying more tumor subtypes. Moreover, the current diagnostic CNN was based on endoscopic forceps biopsy, training and prediction discrepancies may exist between forceps biopsy and resected specimens. Finally, the current SRS system is large and non-portable, future work will focus on the miniaturized and portable instrument for clinical translation[18], even toward the endoscopic SRS to integrate with the current gastroscopy devices[47–49].

In summary, we have developed a U-Net based femto-SRS imaging technique with improved speed and preserved chemical

specificity, capable of rapid histological imaging on fresh gastric tissues. Further combined with CNN, femto-SRH enabled not only automated diagnosis with high accuracy, but also semantic segmentation to reveal the heterogeneous distribution of diagnostic groups and tumor subtypes. These capabilities provide promise in near real-time intraoperative diagnosis on gastroscopic biopsy and evaluation of resection margins of endoscopic dissections.

## Methods

**Patients and tissue specimens**. The study was conducted in accordance with the Declaration of Helsinki, and the protocol was approved by the Ethics Review Board of the Zhongshan hospital (B2021-122R2). Written informed consent was obtained from each patient. The patients were included if they met the criteria: (1) 18–70 years of age; (2) American Society of Anesthesiologists class 1–3; (3) single stomach lesion. The exclusion criteria were: (1) history of allergies; (2) previous abdominal surgery; (3) pregnant woman.

Fresh gastroscopic biopsies were taken by endoscopic forceps (Alton, AF-D2416BTC, China) with typical size of ~2 × 2 mm². Surgical tissues from ESD specimens were obtained at three sites/areas with an approximate size of 2 × 2 mm²: (1) intra-tumor; (2) tumor margin; (3) adjacent normal area ~8 mm away from the margin. The freshly excised tissues were transported to our microscopy lab within 2 h, sandwiched between the glass slide and coverslip with a 200 μm spacer, and imaged immediately by SRS with ~50 s without additional processing. The same specimens were then transported to the pathology department for routine histopathologic analysis.

**Pico/femto dual-mode SRS microscope**. In order to generate pico-SRS and femto-SRS on the same field of view (FOV) for training the deep learning algorithms, we added the femto-SRS optical path to the conventional pico-SRS part (Fig. S1). A single femtosecond laser (OPO, Insight DS + , Newport) with a fixed Stokes beam (1040 nm, ~200 fs) and a tunable pump beam (680–1300 nm, ~150 fs) was used for the dual-mode SRS. Each beam was split into two, sending to the corresponding pico and femto parts of the setup.

The pico-SRS system functioned under the "spectral focusing" mode, both the pump and Stokes beams were linearly chirped to picoseconds through SF57 glass rods[23,26]. The Stokes beam was modulated by an electro-optical modulator at 20 MHz and combined with the pump beam using a dichroic mirror (DMSP1000, Thorlabs). The collinearly overlapped beams were tightly focused onto the sample through the laser scanning microscope (FV1200, Olympus) and an objective lens (UPLSAPO 25XWMP2, NA 1.0 water, Olympus). The transmitted stimulated Raman loss signal was detected by a Si photodiode demodulated by a lock-in amplifier (HF2LI, Zurich Instruments), and generated SRS images as the laser spot raster scanned across the sample. The target Raman frequency was selected by adjusting the time-delay between the two pulses. For histological imaging, we imaged at the two delay positions corresponded to 2845 cm⁻¹ and 2930 cm⁻¹, respectively. Then the raw images were decomposed into the distributions of lipid and protein using the numerical algorithm to yield two-color SRS images[17,26]. Laser powers at the sample were kept as: pump 50 mW and Stokes 50 mW.

In the femto-SRS part, the beams simply by-passed the glass rods and traveled through the rest of the optical components, maintaining femtosecond pulse width and generated femto-SRS images for the same FOVs. The splitting and recombination of the pico- and femto- paths were realized by a polarizing beam splitter, and a quarter-wave plate before the retro-reflection mirror for each beam. Shutters were used to switch between the two modes. Femtosecond laser powers at the sample were kept as: pump 20 mW and Stokes 20 mW, which were below the photodamage threshold of femto-SRS imaging for live cells[31]. In addition, SHG signal was simultaneously harvested using a narrow band pass filter (FF01-405/10-25, Semrock) and a photomultiplier tube in the epi mode. All the images used the same setting of 512 × 512 pixels with a pixel dwell time of 2 μs. To image a large area of tissue, mosaicing and stitching were performed to merge the small FOVs into a large flattened image.

**Deep U-Net for femto- to pico-SRS image conversion**. The neural network for the conversion of femto-SRS to pico-SRS was realized based on U-Net[50]. Python language (https://www.python.org) and Pytorch deep learning framework (https://pytorch.org) were used to build the U-Net. The network was composed of a full convolution layer, five down-sampling layers and five up-sampling layers with a pooling layer in each. The optimizer uses 'RMSprop' optimizer with the following parameters: learning rate = 0.001, weight decay = 1e−8, momentum = 0.9. and 'BCEWithLogitsLoss' was chosen as the loss function. We took the single-channel femto-SRS images and dual-channel pico-SRS images of the same FOVs for training and validation. Femto-SRS images were taken at the zero time-delay between pump and Stokes pulses, whereas the pico-SRS images were taken at 2845 cm⁻¹ and 2930 cm⁻¹ Raman frequencies. 50 groups of femto/pico-SRS images of HeLa cells and 100 groups of gastric tissues were collected. The size ratio between training and test datasets was set to 4:1. The ability of U-Net to split single femto-SRS image to dual-channel pico-SRS images was verified by the minimum difference between the converted and measured pico-SRS images (ground truth).

**Coloring of SRH images**. Conventional coloring of multi-color SRS images uses RGB colormap to represent collagen, lipids and protein, extracted from SHG channel and SRS images of 2845 cm⁻¹ and 2930 cm⁻¹. To generate virtual H&E like SRH images, color conversion was conducted. For the lipid channel, we select a similar to eosin color of the lookup table to map it to pink; for protein gray-scale image, we select a similar to hematoxylin color of the lookup table to map it to dark purple; for collagen, we select the common pathological orange to map. It should be noted that the color projections are all linear mapping. After the three individual channels were color converted, they were merged to form an SRH image similar to H&E. Femto-SRS images were converted to pico-SRS of lipids and protein, and then false colored to femto-SRH images for demonstration, grading and further training.

**Convolutional neural network for histopathological diagnosis**. To diagnose based on femto-SRS with deep learning algorithms and classify the results into different subtypes, we designed two layers of networks to: (1) classify non-cancer vs. cancer; (2) further classify differentiated vs. undifferentiated cancers within the cancer group according to the Japanese Classification of Gastric Carcinoma (JCGC)[51]. The "differentiated cancer" includes well or moderately differentiated papillary adenocarcinoma and tubular adenocarcinoma. And the "undifferentiated cancer" includes poorly differentiated adenocarcinoma, mucinous adenocarcinoma and signet ring cell carcinoma. Both the neural networks were built based on Inception-Resnet-V2 structure[52], using Python language and Pytorch deep learning framework. Inception-Resnet-V2 is a special type of CNN consisted of the concept layer, the ResNet layers and the full connection layers of 1536 units with a depth of 572, including the following sequence: (1) input layer; (2) stem layer; (3) Inception-ResNet-A; (4) Reduction-A; (5) Inception-Resnet-B; (6) Reduction-B; (7) Inception-Resnet-C; (8) average pooling layer; (9) full connection layer.

Gastroscopic tissues from 279 patients were imaged with femto-SRS, and the U-Net converted multi-channel SRS images were used for training the CNN. In the current work, all the CNN classifications were performed on multi-color SRS images, while the corresponding SRH images were mainly used for visual validation by pathologists. The SRS images were annotated by 2 experienced pathologists (ZS Hospital) based on the clinical histopathology result of each patient case, as the "ground truth" for the whole piece of biopsy. The macroscopic and common histological types were defined according to the JCGC classification[51].

The image dataset was randomly divided into training/validation cohort and test cohort with a ratio of 7:3. The training/validation cohort was further divided into 5 equal-sized portions, 1 of which was used as validation data and the other 4 as training data. The choice of validation dataset was rotated among the 5 portions, resulting in fivefold cross validation. Before feeding into the network, the input images were normalized by a Z-Score with the mean of (0.485, 0.456, 0.406) and the standard deviation of (0.229, 0.224, 0.225). The full-sized images were sliced into tiles of 300 × 300 pixels each. The optimizer used 'Adam' optimizer with the following parameters: learning rate = 0.001, betas = (0.9, 0.999), eps = 1e−8, weight_decay = 1e−4. The scheduler used 'MultiStepLR' scheduler with the following parameters: milestones = [200,400,500], gamma = 0.5. And 'CrossEntropyLoss' was chosen as the loss function.

Patient-level diagnosis with deep-learning models was performed in two stages. Each test image was sliced into small tiles of the same size as the training tiles. The first CNN predicted on each tile as normal or tumor, and the tissue-level classification was made by calculating the percentage of tiles that were predicted as cancer, dividing all the test dataset into groups of "non-cancer" and "cancer". The second CNN further classified the "cancer" group into subtypes of "differentiated" and "undifferentiated" following the same method. The optimum thresholds were determined in the training cohort using Youden's index (Fig. S5), indicating a percentage threshold of 0.476 for cancer vs. non-cancer, and 0.505 for differentiated vs. undifferentiated.

**Semantic segmentation and prediction heatmaps**. Probability heatmaps generated prediction probabilities of the classified subtypes and spatial heterogeneity of histological grades within each imaged tissue. High-resolution semantic segmentation was performed as follows:

1. The full femto-SRS image was sliced to tiles of size 300 × 300 pixels, and the image was extended by a margin of one tile's width, flipped from the edge tiles.
2. Each tile was fed into the first CNN, generating output probabilities (between 0 and 1) of non-cancer and cancer with the sum of 1. The tile was classified as non-cancer if the output probability of non-cancer was above 40%, classified as cancer if the probability of cancer was above 40%, and otherwise classified as non-diagnostic. The model classification was insensitive to the threshold in the range of 0.25–0.4.
3. For the tiles classified as cancer, the second CNN was applied for further subtyping. If the differentiated-cancer probability was greater than the undifferentiated, it was classified as differentiated, otherwise it was classified as undifferentiated.
4. Shift the intercepted area to the right by 50 pixels, and repeat steps 2–3 to run the classification. If it had been moved to the edge, shift down 50 pixels and continue moving in the reversed direction.

5. Each sub-tile of $50 \times 50$ pixels within the original image was classified 36 times. Count the times it was classified into each subtype (normal, low-grade, high-grade and non-diagnostic), and divide by 36 to represent the probability of prediction for the subtypes.

6. Remove the extended margin, and recover the image to the original size. Then the probability of each subtype was color-coded and superimposed onto the femto-SRH image to generate the heatmap.

**Comparison between femto-SRH plus CNN and pathologists**. Each fresh gastroscopic biopsy was first imaged with femto-SRS without any processing, and then the tissue was sent for H&E staining. The SRH images were rated by CNNs. Four experienced pathologists from four different hospitals (WH, TX, SR, and XM) were asked to review the H&E images from the testing cohort and give their independent pathological judgment without knowing any clinical information or pathological results. The results were then compared with those of the CNN-predictions on SRH images of the corresponding specimens.

**Statistics and reproducibility**. The ROC, the AUC, accuracy, specificity (Sp), sensitivity (Sn), positive predictive value, and negative predictive value were calculated to evaluate the performance of the CNNs. For each pathologist, we calculated Cohen's kappa statistic to evaluate the level of diagnostic agreement between femto-SRH and H&E results. Cohen's kappa was also calculated for CNN based SRH vs. ground truth (traditional pathology of the same tissue). The McNemar test or Chi-square was applied to compare the diagnostic results. Two-side statistical tests were conducted and the $p$ value was regarded as statistical significance. The analyses were run on SPSS (version 9.0) and R software for Windows (version 3.5.1; http://www.r-project.org). Figures 1c; 3a–d; 4a–e are representative of twenty independent experiments. Figures 2a, c–d; 6b; are representative of five independent experiments. Figure 7b is representative of two independent experiments.

**Reporting summary**. Further information on research design is available in the Nature Research Reporting Summary linked to this article.

## Data availability

The data that support the findings of this study are provided in Supplementary Movies S1–S2, Supplementary Tables S1–S3 and Supplementary Figs. S1–S9. Data used for training and test the deep-learning models are available under open access at [https://zenodo.org/record/6582765]. Source data are provided with this paper.

## Code availability

Source code, comprehensive documentation, use-case tutorials and hyperparameters to perform U-Net and CNN based results presented in this manuscript can be found at https://zenodo.org/record/6582765 under open access.

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

## Acknowledgements

M.J. acknowledges the financial support from the National Key R&D Program of China (2021YFF0502900), the National Natural Science Foundation of China (61975033) and Shanghai Municipal Science and Technology Major Project (2017SHZDZX01, 2018SHZDZX01) and ZJLab. H.H. and P.Z. acknowledge financial support from the National Natural Science Foundation of China (81900548) and the Smart Medical Program of Shanghai Municipal Health Commission (2018ZHYL0204). The pathological study was supported by the following pathologists, to whom we express our deepest gratitude: Dr. Xiangjie Sun (Fudan University Shanghai Cancer Center), Dr. Guojian Gu (1st People's Hospital of Taicang, TC), Dr. Sinian Huang (Zhongshan Qingpu Hospital), Dongxian Jiang (Zhongshan Hospital, ZS), Dr. Jun Zhou (Shangrao Municipal Hospital, SR), and Dr. Lanying Lei (Central Hospital of Wuhan, WH).

## Author contributions

M.J., H.H., and P.Z. designed and directed the study; H.H., W.S., M.W., Q.J., H.G., S.L., J.N., X.Y., and X.G. prepared the tissue samples; Z.L. and J.A. performed stimulated Raman scattering microscopy experiments; Z.L. constructed deep learning algorithms and performed trainings; J.H. performed pathology work; M.J., H.H., and Z.L. wrote the paper with contributions from all the authors.

## Competing interests

The authors declare no competing interests.
