## [Peer Review File · Nature Communications]

Instant diagnosis of gastroscopic biopsy via deep-learned single-shot femtosecond stimulated Raman histologyREVIEWER COMMENTS

Reviewer #1 (Remarks to the Author): expert in deep learning for imaging

Liu et.al. presented a study of using single color femto-SRS to generate H&E images for accurate diagnosis of gastric cancer. Specifically, they show a deep learning model that is able to reproduce the relevant protein and lipid images for stimulated Raman histology from fast-yet-low-spectral-resolution femtosecond SRS images. They then train two deep learning classifiers that predict whether a biopsy sample is normal or cancer and if a cancer sample is differentiated or undifferentiated. The models shown demonstrate high accuracy (>96%) over a set of 279 patients. The authors also demonstrate some interpretability of their models by showing semantic segmentation/heatmaps of biopsies showcasing the inherent heterogeneity of tissue samples, they suggest this can be used to guide tumor resection. There are two main innovations: 1. generating SRH image from a single low spectral resolution SRS image using a U-Net which affords significant speed up of the imaging process. 2. deep learning based diagnosis of gastric cancer. Although the authors have used a similar deep learning approach (reference 16) for laryngeal cancer, the new application in gastric cancer is still worth noting considering the large number of cases studies and the high diagnostic accuracy. If validated in a larger cohort of patients, it could be a powerful tool for future clinical application. The manuscript can be published if a few concerns are addressed.

Comments:

1. In the examples shown, the predicted SRH quality seem to match well with two-color SRS. Is there a quantitative measure for different cases? In previous studies (e.g. Shin, Scientific Reports, 2019, 9, 20392), the SRH can be highly variable depending on tissue composition. In some cases (low lipid or high collagen), the nuclei contrast can be quite low. Does the prediction always give robust results, even when tissue composition changes significantly? It is possible that gastric cancer has more consistent chemical composition, but it would be helpful to see the full range of scenarios.
2. The percentage-tile thresholds used for determining whether a sample was cancer or not cancer is set to maximize model accuracy. This then supposes that if a field of view has less than 47% of its tiles predicted as cancer it is considered normal. However, this seems perhaps arbitrarily high, or calls into question the model's real accuracy. In a pathology setting, if several areas of the sample are identified to be cancer, then the sample would likely to be classified as cancer. The use of a high threshold seems to create an error "buffer" for the model to be incorrect on a given tile but correct in the aggregate so long as an appropriate arbitrary threshold is chosen. Is this due to the limited tile size which increased classification error for individual tiles? In a non-cancerous patient, are those predicted cancer tiles appear cancerous to the pathologists? If not, what does that mean for the accuracy of the model?
3. Since the ResNet can take any images and make a prediction, is the SRH generation step necessary for diagnosis? In other words, if the femto-SRS images are used directly for training the diagnostic model, what would be the accuracy? SRH is helpful as a visual validation, but it may not be necessary for the model.
4. How much role does collagen play in the model accuracy?
5. The semantic segmentation approach is quite interesting, especially for margin analysis. A few example images are shown. However, for non-experts in pathology, it is difficult to see if individual areas are classified properly. To validate the classification of regions, a pathologist should evaluate the same image (or corresponding H&E if possible) and map out those regions. Currently, there is no way of evaluating the accuracy of such segmentation.
6. In the semantic segmentation/prediction heatmaps, the authors imply that tiles with predictions < 40% in both elements, or > 40% in both elements are considered non-diagnostic. The procedure is somewhat unclear to me. The tile prediction gives binary results so where does the probability come from? How are the 40% threshold chosen? Is it based on the previous training?
7. Figure 7B has a related challenge. While the margin guidance is helpful, for a non-expert, similar lines could be possibly drawn in intra-tumor and normal cases. It seems like a threshold is again needed to determine the margin. Is this probability threshold empirically determined from the 5 cases or from previous data? The statistics is less robust with only 5 patients. How is the accuracy determined?

Other comments:

8. Some of the descriptions of pathological features are not consistent with the images shown or not clearly indicated. For example, the authors claim that Figure 3D shows infiltration of small and

dense inflammatory cells, but the SRH image does not reflect those cells (as compared to the H&E). The description of Fig. 3C is somewhat unclear. Is the arrow pointing at a large cell and the mucus region is above the cell? Cells in H&E appeared much smaller and denser.

9. What is the wavelength used? I expect that would affect the contrast of the image and the resulting conversion to SRH. It seems to target lipid transition (at low resolution) as the cellular features are still quite visible. In addition, how much signal gain for femto-SRS vs pico-SRS?

10. We implore the authors to publish their code in a publicly available repository like GitHub for others to use. Whether pre-trained, or ready for training, more available code facilitates growth of the field.

Reviewer #2 (Remarks to the Author): expertise in Raman imaging

The manuscript by Liu et. al demonstrated an instant diagnosis of biopsy using label-free chemical imaging by U-Net-trained femtosecond stimulated Raman histology. The major innovation lies in the application of a U-Net to transform a single-color broadband femto-SRS image into two-color pico-SRS images at protein and lipid-rich regions to form stimulated Raman histology images. Second harmonic generation (SHG) was acquired simultaneously to provide contrast for collagen fibers. A second neural network was then used on the composite images for the classification of cancerous and benign regions. The manuscript is well written and the data represents the claims in the manuscript. However, compared to previous papers published by the same group (ref. 20) and others (e.g., refs. 18, 21), the major innovation is the application of U-net to map femto-SRS into two-color pico-SRS. The same idea has been demonstrated in ref. 30 to map intracellular organelles with femtosecond SRS images. Further, the speed improvement using U-Net & femto-SRS combination is limited, as the ground truth pico-SRS images are only two-color and can be acquired already at high throughput. Considering the efforts and time into training sample acquisition and model training, the U-net & femto-SRS approach might not deliver an overall improvement on the diagnosis throughput. Therefore, I do not recommend the publication of this manuscript on Nat. Comms.

Below are some additional comments:

1. From another paper on ref. 18, a composite of CH₂ (2850 cm⁻¹) and CH₃-CH₂ (2930 – 2850 cm⁻¹) shows better analogy to H&E results, can the authors perform a comparison between this approach and the ones used in the manuscript?

2. In many example images, I do not see significant contributions from collagen. Does that mean collagen is not a significant feature in H&E staining and histological analysis?

3. In the manuscript the authors mentioned that one advantage of femto-SRS is the higher SNR than pico-SRS, while this holds true under the same laser powers, femto-SRS is known to have a lower laser damage threshold, which limits the effective SNR we can achieve under the same laser powers. This limitation again hinders the idea of femto-SRS & U-Net.

Reviewer #3 (Remarks to the Author): expertise in gastric cancer

This is a very impressive study and a well written manuscript. The ability to provide rapid microscopic AI-assisted diagnosis for gastric biopsies would be a significant clinical improvement to the current standard of histopathologic processing and interpretation. The results are extremely impressive and support the claims. The methodology appears to be sound. It will be interesting to see if this technology translates into the clinical workflow.

We thank the reviewers for their constructive comments, based on which we have revised our manuscript. All the changes have been made in the revised manuscript with highlighted markups. Below are the point-by-point responses to all the reviewers' comments:

Reviewer #1:

Liu et.al. presented a study of using single color femto-SRS to generate H&E images for accurate diagnosis of gastric cancer. Specifically, they show a deep learning model that is able to reproduce the relevant protein and lipid images for stimulated Raman histology from fast-yet-low-spectral-resolution femtosecond SRS images. They then train two deep learning classifiers that predict whether a biopsy sample is normal or cancer and if a cancer sample is differentiated or undifferentiated. The models shown demonstrate high accuracy (>96%) over a set of 279 patients. The authors also demonstrate some interpretability of their models by showing semantic segmentation/heatmaps of biopsies showcasing the inherent heterogeneity of tissue samples, they suggest this can be used to guide tumor resection. There are two main innovations: 1. generating SRH image from a single low spectral resolution SRS image using a U-Net which affords significant speed up of the imaging process. 2. Deep learning based diagnosis of gastric cancer. Although the authors have used a similar deep learning approach (reference 16) for laryngeal cancer, the new application in gastric cancer is still worth noting considering the large number of cases studies and the high diagnostic accuracy. If validated in a larger cohort of patients, it could be a powerful tool for future clinical application. The manuscript can be published if a few concerns are addressed.

Response: We truly appreciate the reviewer for the careful review and positive comments.

Comments:

1. In the examples shown, the predicted SRH quality seem to match well with two-color SRS. Is there a quantitative measure for different cases? In previous studies (e.g. Shin, Scientific Reports, 2019, 9, 20392), the SRH can be highly variable depending on tissue composition. In some cases (low lipid or high collagen), the nuclei contrast can be quite low. Does the prediction always give robust results, even when tissue composition changes significantly? It is possible that gastric cancer has more consistent chemical composition, but it would be helpful to see the full range of scenarios.

Response: We thank the reviewer for bringing up this important issue. The quality of SRH in our work depends on two main aspects. First, the prediction accuracy of the U-Net across different tissue conditions. This could be optimized by including as many types of tissues as possible in the training dataset. As shown in our results (Fig. 2 and Fig. S2), the quantitative agreements between predicted and ground truth SRS are high. And from Fig. 3, Fig. 4 and Fig. S3, we can see different conditions of gastric tissues (normal, different grades of tumor, and different types of tissue) showed consistent predicted SRS results. Second, the nuclei contrast indeed depends on the relative compositions of lipid and protein. In general, high content of extracellular lipids may results in high contrast of nuclei structures (such as in normal brain tissues, ref. 14, 15,

18), but too much content of intracellular lipids tends to degrade nuclei contrast (such as in adipocytes). And as indicated by Shin et.al, in the extreme cases of low lipid or high collagen, the nuclei contrasts are also low (such as in meningioma – a type of brain tumor). In gastric tissues, these extreme cases are quite rare. We have added the quantitative measurement of the lipid/protein ratios for all the 279 cases as shown in Fig. S9a, indicating the majority (80%) of these gastric tissues fall into a relatively narrow ratio range between 0.62-0.86, which ensured high consistency of SRS/SRH quality that covered a broad range of gastric tissue types (as shown in Figs. 3-4, S3, and Movie S1-S2). We also quantified the area ratio of collagen fibers in all the cases, and showed that the extreme cases of high collagen content were very rare (Fig. S9b). Nonetheless, note that our applied SRH coloring method was not deep-learning based, which suffered from color adjusting and tweaking on a case-by-case basis. The reviewer was right that in the cases of tissues with high heterogeneity and abrupt changes of composition (such as in breast tissues with adipocytes and dense collagen fibers), such method will become quite challenging to retain high consistency. And in those cases, other types of deep-learning algorithms may be more suited. We are currently working on developing new methods for SRH color mapping to match H&E with much improved quality. We have added discussions on these issues in page 13, paragraph 1, and included related references 45 and 46.

Fig. S9. Histograms of quantitative biochemical compositions. (a) Lipid/protein ratio; (b) Collagen area ratio.

2. The percentage-tile thresholds used for determining whether a sample was cancer or not cancer is set to maximize model accuracy. This then supposes that if a field of view has less than 47% of its tiles predicted as cancer it is considered normal. However, this seems perhaps arbitrarily high, or calls into question the model's real accuracy. In a pathology setting, if several areas of the sample are identified to be cancer, then the sample would likely to be classified as cancer. The use of a high threshold seems to create an error "buffer" for the model to be incorrect on a given tile but correct in the aggregate so long as an appropriate arbitrary threshold is chosen. Is this due to the limited tile size which increased classification error for individual tiles? In a non-cancerous patient, are those predicted cancer tiles appear cancerous to the pathologists? If not, what does that mean for the accuracy of the model?

Response: We thank the reviewer for the critical comments. It is true that in a pathology setting, only a few sites of tissue identified as cancer is enough to rate the whole sample as cancer, because pathologists have high diagnostic accuracy on high-resolution details, including cellular and subcellular features. However, the current model could not work in the same way as pathologists do. The classification error for individual tile may be caused by a few reasons. First, the labeling errors exist in the training dataset. We treated the whole tissue annotated as cancer to be uniformly cancerous for each sliced tile, which introduces mislabeling of certain non-cancer tiles within the same sample due to the intra-tumor heterogeneity. In contrast, treating the whole non-cancer tissue uniformly was much less likely to mislabel cancer tiles. Second, the reviewer was right that the limited tile size also affected the prediction accuracy. Therefore, there exist higher probability of prediction errors on non-cancer tiles than cancer ones. In the mis-predicted non-cancer cases (Fig. S8), the tiles predicted cancer did appeared abnormal, which were more likely due to tissue degradation (Fig. S8a) or dense macrophage cells in inflammatory tissues (Fig. S8b). Moreover, the distribution pattern of the percentage-tiles (Fig. 5B) showed that most cancer cases had percentage-tile higher than 75%, and most non-cancer cases were lower than 25%. Considering these factors, setting a seemingly high threshold around 50% was effective to increase the prediction accuracy at the patient-level, similar to the methodology used by ref. 18. The optimum threshold of 47% was determined by Youden's index. In order to reach high accuracy at the tile-level, it requires high-precision labeling for each tile in the training dataset, which could not be achieved with the current model on fresh tissues images, because the corresponding H&E images could not be generated.

For some types of tumors, such as breast cancer, it is possible that only small portion of the tissue is diagnostic, while the rest of the tissue might be the non-diagnostic (collagen fibers or fat tissues). In these situations, the current ResNet and percentage-tile based model would not be suited, and we are working on applying weakly-supervised and multi-instance learning to solve the problem.

We have added discussion on these issue in page 13, paragraph 1.

Fig. S8. Demonstration of the two misclassified non-cancer cases. (a) case #54 with degraded tissues; (b) case #186 with dense macrophage-like cells. Red squares: predicted cancer; cyan squares: predicted non-cancer.

3. Since the ResNet can take any images and make a prediction, is the SRH generation step necessary for diagnosis? In other words, if the femto-SRS images are used directly for training the diagnostic model, what would be the accuracy? SRH is helpful as a visual validation, but it may not be necessary for the model.

Response: We regret for making the confusion. In fact, we did use U-net converted multi-color femto-SRS images for training the diagnostic model (as explained in “Methods - Convolutional neural network for histopathological diagnosis” section). The reviewer was right that SRH was unnecessary for model training and prediction, it was mainly used for visual validation. Since the optimization of SRH for each image requires fine tuning of the color balance, it would be extremely low efficient to generate SRH images for all the 279 cases, and might yield additional variations during the process. We have corrected the typo in page 7, paragraph 3. “...femto-SRH images from 279 patients...” was revised to “...femto-SRS images from 279 patients...”.

4. How much role does collagen play in the model accuracy?

Response: We thank the reviewer for bringing up the important issue. For pathologists, collagen may not be the key information for cancer diagnosis, the most critical

diagnostic features are cellular morphology, nuclei morphology and tissue patterns, etc. For pure diagnostic purpose, it might not be necessary to include collagen in the model. However, we notice that collagen content is correlated with tumor structures and metastasis (ref. 41-43), and our results showed that cancerous tissues have slightly more collagen contents than non-cancer cases (Fig. S9b). Therefore, for offering more intact imaging details, we preferred to keep the collagen channel.

We have performed the same model without collagen (SHG) channel for comparison, and found the model accuracy remains unchanged. For the misclassified non-cancer case with high collagen content (# 186, Fig. S8b), the mis-prediction was not caused by collagen fiber, but mainly the inflammatory tissue condition with dense macrophage-like cells. For future work, more emphasis might be needed to fully explore and utilize the collagen information (such as morphology and aging) to aid diagnosis.

We have added discussions on collagen related topic in page 12, paragraph 1.

5. The semantic segmentation approach is quite interesting, especially for margin analysis. A few example images are shown. However, for non-experts in pathology, it is difficult to see if individual areas are classified properly. To validate the classification of regions, a pathologist should evaluate the same image (or corresponding H&E if possible) and map out those regions. Currently, there is no way of evaluating the accuracy of such segmentation.

Response: We agree with the reviewer that ideally, we should ask pathologists to label different regions and validate the classification accuracy. However, it was limited by a few factors. First, we imaged fresh unprocessed tissues, which was difficult to create corresponding H&E sections to compare with. Second, although the SRH images have similar color schemes as H&E, they are still quite different from true H&E. Pathologists could learn to rate the whole SRH image, but they are not trained to accurately label the fine margins based on SRH contrast. The three representative images (Fig. 6B-C) from different AI-classified regions were indeed validated by pathologists. Lastly, the model accuracy was mainly supported by the previous training and optimization process, although the current model could not guarantee high precision for individual tile, the accuracy for defining each region composed with a group of tiles is expected to be high. The current segmentation approach is more of a proof-of-principle demonstration (similar to ref.21). Ideal work will need to perform SRS and H&E on the same tissue section to generate pixel-level agreement, and use the margin labeled by pathologists as the ground truth to train the neural network. We have added discussions in page 10, paragraph 1.

6. In the semantic segmentation/prediction heatmaps, the authors imply that tiles with predictions < 40% in both elements, or > 40% in both elements are considered non-diagnostic. The procedure is somewhat unclear to me. The tile prediction gives binary results so where does the probability come from? How are the 40% threshold chosen? Is it based on the previous training?

Response: We would like to clarify the confusion. The non-diagnostic tiles were mainly originated from blank areas without any tissue or with only small portion of tissue. The

output probability (generated by SoftMax) of each tile was not binary, but a value between 0 and 1 for each element (e.g. cancer or non-cancer). For the tiles with obvious tissue morphology, the output value would be most likely close to 0 or 1, whereas for the non-diagnostic tiles, the values were closer to 0.5. Therefore, we empirically chose the range of 0-0.4 as non-cancer, 0.6-1 as cancer, and 0.4-0.6 as non-diagnostic, based on previous trainings. We found that the results were insensitive to the threshold, changing 0.4-0.6 to 0.3-0.7 did not make much difference. For the whole-tissue diagnosis, the percentage of tiles were calculated without the non-diagnostic tiles, taking only the cancer and non-cancer tiles into account. And note that for the subtyping of differentiated vs. undifferentiated within the cancer group, the non-diagnostic tiles did not exist. Therefore, as shown in Fig. 6A, four total categories were generated for semantic segmentation. We have revised the Methods to clarify these issues (page 18, paragraph 3).

7. Figure 7B has a related challenge. While the margin guidance is helpful, for a non-expert, similar lines could be possibly drawn in intra-tumor and normal cases. It seems like a threshold is again needed to determine the margin. Is this probability threshold empirically determined from the 5 cases or from previous data? The statistics is less robust with only 5 patients. How is the accuracy determined?

Response: Similar to comment 6, the segmentation results were insensitive to the threshold, which was empirically determined from the previous training data, not the 5 test cases. The ESD cases were not easily accessible, and the 5 cases studied in this work were mainly for demonstration purpose of our method. The dashed margin was mainly for visual illustration, with the cancer and non-cancer areas validated by a pathologist. It is true that histological heterogeneities remain in the intra-tumor and normal specimens, and tumor margin is not a clear and solid interface. But we can still see the clear trend of increasing cancer probability from normal to intra-tumor regions. Large cohort size will be needed for robust statistics, which is an on-going work that takes much longer time.

8. Some of the descriptions of pathological features are not consistent with the images shown or not clearly indicated. For example, the authors claim that Figure 3D shows infiltration of small and dense inflammatory cells, but the SRH image does not reflect those cells (as compared to the H&E). The description of Fig. 3C is somewhat unclear. Is the arrow pointing at a large cell and the mucus region is above the cell? Cells in H&E appeared much smaller and denser.

Response: We thank the reviewer for pointing out these issues. For Fig. 3D, the small and dense inflammatory cells were not clearly visualized due to the small cell size and relative high-protein low-lipid content in the surrounding matrix, hence the contrast of cellular boundaries was relatively low. For Fig. 3C, the arrows were pointing at the large cells (so called cup-shaped cells), not the mucus region, we have revised the figure caption to clarify this. Certain cells in H&E appeared denser and smaller due to the dehydration process. As explained in the revised discussion in page 7, paragraph 2, the different appearances between SRS and H&E were mainly caused by differences in

biochemical contrast, tissue processing and staining variations, etc.

9. What is the wavelength used? I expect that would affect the contrast of the image and the resulting conversion to SRH. It seems to target lipid transition (at low resolution) as the cellular features are still quite visible. In addition, how much signal gain for femto-SRS vs pico-SRS?

Response: In our instrument, the pump and Stokes beams were set at 802 nm and 1040 nm, respectively. Given the broad spectral width of the femtosecond pulses ($\sim 150 \text{ cm}^{-1}$), a single femto-SRS image covers the Raman frequency range of 2800-2950 cm^{-1} , which included both the lipid and protein contents. And on the basis of the included information, we recovered the image channels of lipid and protein using deep U-Net. For the same laser powers of both pump and Stokes beams, our measured signal gain is ~ 6 for femto-SRS over pico-SRS. And the actual applied femtosecond laser power was 40% that of the picosecond version.

10. We implore the authors to publish their code in a publicly available repository like GitHub for others to use. Whether pre-trained, or ready for training, more available code facilitates growth of the field.

Response: We thank the review for the suggestion, we totally agree to share our codes in the public repository.

Reviewer #2:

The manuscript by Liu et. al demonstrated an instant diagnosis of biopsy using label-free chemical imaging by U-Net-trained femtosecond stimulated Raman histology. The major innovation lies in the application of a U-Net to transform a single-color broadband femto-SRS image into two-color pico-SRS images at protein and lipid-rich regions to form stimulated Raman histology images. Second harmonic generation (SHG) was acquired simultaneously to provide contrast for collagen fibers. A second neural network was then used on the composite images for the classification of cancerous and benign regions. The manuscript is well written and the data represents the claims in the manuscript. However, compared to previous papers published by the same group (ref. 20) and others (e.g., refs. 18, 21), the major innovation is the application of U-net to map femto-SRS into two-color pico-SRS. The same idea has been demonstrated in ref. 30 to map intracellular organelles with femtosecond SRS images. Further, the speed improvement using U-Net & femto-SRS combination is limited, as the ground truth pico-SRS images are only two-color and can be acquired already at high throughput. Considering the efforts and time into training sample acquisition and model training, the U-net & femto-SRS approach might not deliver an overall improvement on the diagnosis throughput. Therefore, I do not recommend the publication of this manuscript on Nat. Comms.

Response: We greatly appreciate the reviewer's constructive comments and critiques. (1) Regarding the novelty of our work. It is true that ref.30 used a similar idea of applying deep-learning algorithm to classify different groups of intracellular organelles in living cells, with femtosecond SRS served as the input data, hyper-spectral SRS and

spectral phasor classified organelles as the ground-truth data. Also, a related work in ref.32 applied deep-learning to extract the full SRS spectra from femtosecond SRS, and did an excellent job in differentiating various molecular compositions. For comparison, these former studies required the full spectral information as the training data, while in our work the full-spectrum is unnecessary for rapid diagnosis, but rather the two-channel pico-SRS is sufficient for histology with much improved efficiency. Therefore, our method was optimized specifically for stimulated Raman histology. Although there have been numerous works of SRS histology, we believe our work was the first attempt to convert single-shot femto-SRS to SRH for tissue histology and disease diagnosis. In addition, compared with previous work (eg. ref. 18, 20, 21), our work is the first comprehensive demonstration for gastric cancer, particularly compatible with the setting of gastroscopy, which has critical importance in early gastric cancer screening. The urgent need of rapid diagnosis and the small size of biopsy may enable the work to be a potential killer application of SRS microscopy in biomedicine and clinical translation.

(2) Regarding the speed improvement of our method. Even for two-color SRS, the best result of pico-SRS in imaging $2 \times 2 \text{ mm}^2$ tissue was about 2 minutes (ref. 21), whereas in our single-shot femto-SRS the time cost had been reduced to less than 1 minute, and it could be easily reduced to $<30\text{s}$ with optimized imaging and mosaicking methods ($\sim 450 \mu\text{m}$ FOV, ~ 25 image frames, typical $1\text{s}/\text{frame}$), which will be about ~ 4 times improvement in imaging speed. Given the short examination time in gastroscopy, such an improvement in intraoperative histology is critical to minimize the perturbation on ordinary gastroscopic process. The reviewer was right that advanced techniques could achieve rapid multi-color imaging as shown in previous works (ref. 26, 27, 29), but they require complex engineering of the optical design. Femto-SRS is so far the simplest in optical setup, and more importantly, the most stable in signal generation, since there is no need to scan the laser wavelength or time-delay. Such simplicity and stability will be critical for long-term imaging in clinical translation. Furthermore, the training data acquisition and training time shall not be the limiting issues, the training and validation time of the 224 cases (70740 title) was ~ 115 hours. And as we know, once the training is done, we do not need to retrain the model, unless additional types of disease tissues must be included. Therefore, comparing the engineering efforts and maintaining costs in complex pico-SRS technique to achieve the same speed, we believe the deep-learned femto-SRS approach has sufficient overall advantages tailored for rapid histopathology.

We have added discussions in page 11, paragraph 2.

Below are some additional comments:

1. From another paper on ref. 18, a composite of CH₂ (2850 cm⁻¹) and CH₃-CH₂ (2930 – 2850 cm⁻¹) shows better analogy to H&E results, can the authors perform a comparison between this approach and the ones used in the manuscript?

Response: We thank the reviewer for the insightful observation. In fact, we applied essentially the same method to generate composite SRS and SRH images as ref. 18, which was based on the linear decomposition method we developed in ref. 14 and 15. We believe the major difference was originated from the different tissue types. Brain

tissues are more homogeneous and has better nuclei contrast due to higher extracellular lipid contents. We could achieve the same quality as ref. 18 on fresh brain tissues (Fig. R1), while the coloring for gastric tissues is somewhat different because of the different lipid/protein ratio and tissue morphology. As we noticed, the current SRH coloring method used by different research groups (ref. 18, 45, 46) may not be the optimum method for general tissue histology. Our on-going research is aiming at improving the SRH quality using advanced deep-learning algorithms to generate images much more like traditional histopathology. We have added discussion in page 13, paragraph 1.

Fig. R1. SRH images of a human brain tumor tissue taken at two representative tissue sites (a) and (b). Scale bars: 30 μ m.

2. In many example images, I do not see significant contributions from collagen. Does that mean collagen is not a significant feature in H&E staining and histological analysis?

Response: It is true that collagen contribution is insignificant in most of our studied tissues (Fig. S9b). For pathologists, collagen fiber is not the key histological feature for cancer diagnosis. The main diagnostic features are cellular morphology, nuclei morphology and tissue patterns, etc. However, that does not mean collagen is not related to disease conditions. Collagen content has been found to be correlated with tumor genesis and metastasis (ref.41-43). Our data also showed that cancerous tissues on average have slightly more collagen contents than non-cancer cases (Fig. S9b). In our model, we did find that collagen content has very weak effect on prediction accuracy for gastric tissues (please see our response to Reviewer1, Comment 4). But to keep more intact information, we prefer to include the collagen information in the images. We have added discussions on collagen related topic in page 12, paragraph 1.

3. In the manuscript the authors mentioned that one advantage of femto-SRS is the higher SNR than pico-SRS, while this holds true under the same laser powers, femto-SRS is known to have a lower laser damage threshold, which limits the effective SNR we can achieve under the same laser powers. This limitation again hinders the idea of femto-SRS & U-Net.

Response: We thank the reviewer for the careful thought. We can have a back-of-the-

envelope calculation: in our system, the 200-250 fs pulses generate femto-SRS signal ~ 6 times the intensity of the pico-SRS (1-2 ps). To keep a similar signal level, we reduced the femtosecond laser power of both beams to 40% the powers of the picosecond version, while the shot-noise reduces to ~63% (1.6x the SNR). As informed in the Methods, 20 mW (~2 MW/cm²) of both beams was used in our study, which was much below the damage threshold of femtosecond SRS imaging for live cells (~8 MW/cm²) as investigated by ref.31 for similar wavelengths and pulse durations. Hence there is still plenty room to increase the SNR without photodamage. In addition, compared with long-time live cell imaging, *ex vivo* tissue imaging for diagnostic purpose usually requires only one-time laser scanning, thus laser damage is much less of an issue for tissue imaging.

Reviewer #3:

This is a very impressive study and a well written manuscript. The ability to provide rapid microscopic AI-assisted diagnosis for gastric biopsies would be a significant clinical improvement to the current standard of histopathologic processing and interpretation. The results are extremely impressive and support the claims. The methodology appears to be sound. It will be interesting to see if this technology translates into the clinical workflow.

Response: We very much appreciate the reviewer's positive comments.

REVIEWER COMMENTS

Reviewer #1 (Remarks to the Author):

Liu et.al. have addressed most concerns adequately. One remaining question is the original comment 3 ". Since the ResNet can take any images and make a prediction, is the SRH generation step necessary for diagnosis? In other words, if the femto-SRS images are used directly for training the diagnostic model, what would be the accuracy?" The question is really asking about whether the single femto-SRS image can be directly used for training the diagnostic model instead of using the two-color SRS (converted by U-Net). I would expect the accuracy to be similar because the same input is used. If that is the case, then the conversion to two-color SRH is not very useful (only serve a visualization purpose). Because individual adjustment of SRH to H&E is needed, and diagnosis does not depend on SRH, it seems diagnosis using single femto-SRS image is a more efficient approach.

Reviewer #2 (Remarks to the Author):

The authors have carefully addressed various questions from all the reviewers. I would like to recommend the publication of the manuscript in Nature Communications.

We thank the reviewers for their efforts and insightful comments, based on which we have revised our manuscript. All the changes have been made in the revised manuscript with highlighted markups. Below is the response to reviewer #1's comment:

Reviewer #1:

Liu et.al. have addressed most concerns adequately. One remaining question is the original comment 3 ". Since the ResNet can take any images and make a prediction, is the SRH generation step necessary for diagnosis? In other words, if the femto-SRS images are used directly for training the diagnostic model, what would be the accuracy?" The question is really asking about whether the single femto-SRS image can be directly used for training the diagnostic model instead of using the two-color SRS (converted by U-Net). I would expect the accuracy to be similar because the same input is used. If that is the case, then the conversion to two-color SRH is not very useful (only serve a visualization purpose). Because individual adjustment of SRH to H&E is needed, and diagnosis does not depend on SRH, it seems diagnosis using single femto-SRS image is a more efficient approach.

Response: We very much appreciate the reviewer for bringing up this important issue. It is true that the raw single-channel femto-SRS images can be taken directly to train the same CNN for classification. However, the U-Net conversion step has added flavors of chemical contrast to the original morphology-only data. Such additional information is introduced in the U-Net training process, at the cost of including paired pico-SRS images with chemical resolution. Therefore, the U-Net converted multi-color SRH images should contain more information than the original femto-SRS images. For direct comparison, we have conducted CNN training of the same dataset using the corresponding single-channel femto-SRS data, and the prediction results are indeed underperformed, as shown in Fig. R2. We have added discussions in page 11, paragraph 2 to clarify this issue.

In a related topic of virtual histological staining, single-channel images (such as autofluorescence) were converted to multi-colored H&E virtual stains via deep-learning [*Nat. Biomed. Eng.*, 2019, 3, 466], and reached equivalent diagnosis to true

H&E. This also supports that deep-learning converted images could contain more histological information than the original single-channel data. And our on-going research is aiming at a similar goal with SRS inputs.

Fig. R2. CNN diagnostic predictions using single-channel femto-SRS images. (a-d) Prediction results on test dataset. (e) Comparison between the accuracies using U-Net converted SRH and femto-SRS images.